# Bladder Microbiota Are Associated with Clinical Conditions That Extend beyond the Urinary Tract

**DOI:** 10.3390/microorganisms10050874

**Published:** 2022-04-22

**Authors:** Jan Hrbacek, Vojtech Tlaskal, Pavel Cermak, Vitezslav Hanacek, Roman Zachoval

**Affiliations:** 1Department of Urology, 3rd Faculty of Medicine, Charles University and Thomayer Hospital, 14059 Prague, Czech Republic; vitezslav.hanacek@ftn.cz (V.H.); roman.zachoval@ftn.cz (R.Z.); 2Soil & Water Research Infrastructure, Biology Centre of the Czech Academy of Sciences, 37005 Ceske Budejovice, Czech Republic; vojtech.tlaskal@bc.cas.cz; 3Department of Clinical Microbiology, Thomayer Hospital, 14059 Prague, Czech Republic; cermapav@seznam.cz

**Keywords:** urinary microbiota, aging, diabetes mellitus, dyslipidemia, smoking, erectile dysfunction, chronic kidney disease, antibiotics

## Abstract

Background. Since the discovery of the human urinary microbiota (UM), alterations in microbial community composition have been associated with various genitourinary conditions. The aim of this exploratory study was to examine possible associations of UM with clinical conditions beyond the urinary tract and to test some of the conclusions from previous studies on UM. Methods. Catheterised urine samples from 87 men were collected prior to endoscopic urological interventions under anaesthesia. The composition of the bacterial community in urine was characterized using the hypervariable V4 region of the 16S rRNA gene. Samples from 58 patients yielded a sufficient amount of bacterial DNA for analysis. Alpha diversity measures (number of operational taxonomic units, ACE, iChao2, Shannon and Simpson indices) were compared with the Kruskal–Wallis test. Beta diversity (differences in microbial community composition) was assessed using non-metric dimensional scaling in combination with the Prevalence in Microbiome Analysis algorithm. Results. Differences in bacterial richness and diversity were observed for the following variables: age, diabetes mellitus, dyslipidemia, smoking status and single-dose preoperative antibiotics. Differences in microbial community composition were observed in the presence of chronic kidney disease, lower urinary tract symptoms and antibiotic prophylaxis. Conclusions. UM appears to be associated with certain clinical conditions, including those unrelated to the urinary tract. Further investigation is needed before conclusions can be drawn for diagnostics and treatment.

## 1. Introduction

Since the inception of the Human Microbiome Project (HMP) in 2005 [1], there has been an explosion of knowledge on the composition and some of the roles of the commensal microorganisms residing in and on the human body. Alterations of the microbiota have been associated with the health and disease of the respective organ systems: an association was reported between colonic carcinogenesis and gut microbiota although this link cannot yet be regarded as a causal one [2]. Inflammatory bowel disease (Crohn’s disease and ulcerative colitis) are both associated with reduced complexity of intestinal microbiota, specifically the overgrowth of the phylum *Proteobacteria* and *Enterobacteriaceae* family to the detriment of the phylum *Firmicutes* and the class *Clostridia* in particular [3]. Female lower reproductive tract dysbiosis may play a role in the initiation and progression of carcinogenesis in gynaecological organs [4]. The microbiota of the respiratory tract seems to be a co-factor in the etiology of asthma and cystic fibrosis [3].

Similar to the respiratory tract, the urinary tract was long considered a sterile environment and was not included in the HMP. A decade after bacterial 16S rRNA gene sequences were detected in female urine samples [5] and live microorganisms were cultured using extended quantitative urine culture (EQUC) from samples classified as negative by standard urinary culture techniques [6], the existence of human urinary microbiota (UM) is now widely accepted. Changes in the UM have been observed in association with bladder dysfunction caused by spinal cord injury [7], lower urinary tract symptoms (LUTS) associated with benign prostatic hyperplasia [8], clinical profiles of LUTS in female recurrent urinary tract infections [9], painful conditions of the urinary bladder known as interstitial cystitis [10] and chronic prostatitis/chronic pelvic pain syndrome [11] and even bladder cancer [12].

The interplay between microbiota and the human host will likely turn out to be more complex than it is currently understood. Emerging evidence shows that the gut microbiome may be associated with extraintestinal inflammatory conditions. Increased relative abundance of *Prevotella copri* was detected in patients with new-onset rheumatoid arthritis and mice colonized with *P. copri* suffered from a more serious course of experimentally induced colitis [13]. Intestinal microbiota impacts the balance between pro- and antiinflammatory immune responses in a mouse model of multiple sclerosis, a progressive demyelination disorder of the central nervous system [14]. Anecdotal evidence exists of an epilepsy patient whose seizures ceased after faecal microbiota transplant; the gut microbiome of patients with drug-sensitive epilepsy differed from that of drug-resistant patients [15]. Some studies report differences in gut microbiome composition between patients with depression versus healthy controls but so far, these results have been contradictory [16]. Only a few reports on the links between UM and extraurinary conditions have been published [17,18,19].

We conducted a hypothesis-generating study to explore potential associations between UM and various physiological and pathological conditions and to test some of the putative UM associations reported in previous studies.

## 2. Materials and Methods

### 2.1. Population

Study participants were men undergoing endoscopic urological procedures under anaesthesia in the Department of Urology, 3rd Medical Faculty of Charles University and Thomayer Hospital, Prague, Czech Republic. Inclusion criteria were as follows: a negative result of standard urine culture preoperatively, no foreign body in the urinary bladder (e.g., indwelling catheters, ureteric stents, or bladder stones) and no antibiotic treatment for any medical condition six weeks prior to enrolment. The indications for endoscopic surgery were upper urinary tract stone disease (n = 23), bladder outlet obstruction (n = 8) and urinary bladder cancer (n = 27).

The study was conducted in accordance with the Declaration of Helsinki after previous approval by the Ethics Committee and informed consent was obtained from all participants.

Univariate analyses were performed for specific conditions in which study participants without the respective condition served as the control group. Alpha and beta diversity analyses were performed for age groups (older than 75 years vs. younger); hypertension, diabetes mellitus, chronic kidney disease, dyslipidemia, smoking; the presence of ureteric stents; post-void residual urine (presence vs. absence); body mass index, lower urinary tract symptoms, erectile dysfunction (intergroup comparisons); and “no growth” vs. “insignificant growth” on preoperative urine culture (see Appendix A).

### 2.2. Sample Handling

Catheterised urine specimens were uniformly obtained in the theatre at the beginning of the procedure after disinfection of the genital area, surgical draping and immediately upon endoscope insertion. A water-based jelly without disinfectant (Optilube, Optimum Medical Solutions, Leeds, UK) was used. All samples were stored at 4 °C and frozen on the same day at −20 °C until DNA extraction. No nucleic acid preservatives were used.

### 2.3. DNA Extraction and PCR

The bacterial 16S rRNA gene was extracted from urine samples using the Eligene Urine Isolation Kit (Elisabeth Pharmacon, Brno, Czech Republic) according to the manufacturer’s instructions. DNA extraction controls, as well as negative controls for PCR reactions, were included.

The primers 515F (5′-GTGCCAGCMGCCGCGGTAA) and 806R (5′-GGACTACHVGGGTWTCTAAT) were used to amplify the hypervariable region V4 of the 16S rRNA gene [20]. Each forward primer was barcoded by a sequence of nucleotides designed to multiplex different samples. PCR was performed in triplicates, and every reaction contained 5 μL of 5 × Q5 Reaction Buffer for Q5 High-Fidelity DNA polymerase; 0.25 μL Q5 High-Fidelity DNA polymerase; 5 μL of 5 × Q5 HighGC Enhancer; 1.5 μL of BSA (10 mg mL^−1^); 0.5 μL of PCR Nucleotide Mix (10 mM); 1 μL of primer 515 F (10 μM); 1 μL of primer 806 R (10 μM,); 1.0 μL of template DNA and sterile ddH2O up to 25 μL. Conditions for amplification started at 94 °C for 4 min followed by 25 cycles of 94 °C for 45 s, 50 °C for 60 s, 72 °C for 75 s and finished with a final setting of 72 °C for 10 min.

Three PCR reactions were pooled together, purified by MinElute PCR Purification Kit (Qiagen, Germantown, MD, USA) and mixed in equimolar amounts according to the concentration measured on the Qubit 2.0 Fluorometer (Thermo Fisher Scientific, Waltham, MA, USA). Sequencing libraries were prepared using the TruSeq PCR-Free Kit (Illumina Inc., San Diego, CA, USA) according to the manufacturer’s instructions and sequencing was performed on Illumina MiSeq (2 × 250 bases).

### 2.4. Statistical Analyses

Demographic and clinical data were analysed as continuous or categorical variables and reported as mean and standard deviation (SD) or counts and percentages as appropriate. Age groups were allocated by quartiles; the presence of chronic kidney disease (CKD) was classified by estimated glomerular filtration rate (eGFR) calculated by the CKD-EPI (Chronic Kidney Disease Epidemiology Collaboration) formula; information on diabetic status, hypertension and hypercholesterolemia was extracted from patient medical and drug history. Smoking was classified as current smoker vs. current non-smoker. XLSTAT 2021.2.1 (Addinsoft, New York, NY, USA) was used for statistical calculations. 

The sequencing data were processed using SEED 2.1.05 [21]. Pair-end reads were merged using fastq-join [22]. Sequences with ambiguous bases were omitted as well as sequences with average quality PHRED scores <30. Chimeric sequences were detected and removed using Usearch 8.1.1861 and clustered into OTUs using the uparse algorithm [23] at a 97% similarity level. The most abundant sequence from each cluster [24] was assigned to the closest hits from the SILVA SSU database r138 [25] by DECIPHER 2.18.1 package [26] with a threshold of 40. Sequences identified as non-bacterial were discarded. The DNA sequences have been deposited at the NCBI SRA (PRJNA744742). (See Appendix A for clinical details associated with each sample.)

An elementary description of microbiota on a molecular basis consists of alpha and beta diversity analysis. Alpha diversity relates to the number of specific taxa (here referred to as operational taxonomic unit—OTU) in a sample and the number of individuals within each taxon. Mathematical formulae of commonly used measures take into account sample richness (number of taxons) and evenness (distribution of relative abundances of the taxons), putting more emphasis on one or the other [27,28]. In contrast, beta diversity analyses are mathematical models that quantify the (dis)similarity between microbiome pairs [29]. These methods try to visualize dissimilarity between objects in two- or three-dimensional space.

Analyses of alpha and beta diversity were performed using the packages vegan 2.5-7 [30] and phyloseq 1.34.0 [31] in R 4.0.5 [32]. Singleton OTUs were omitted from beta diversity analysis. Samples with <900 sequences and Good’s coverage <0.85 were excluded. The number of OTUs representing 95% of the community, iChao2 [33], ACE [34], Shannon and Simpson indices were calculated. The Prevalence Interval for Microbiome Evaluation (PIME) algorithm 0.1.0 [35] was used to identify the most relevant OTUs using their prevalence. The PIME algorithm is used to identify the most relevant OTUs to separate groups of samples. The method is based on the concept of prevalence, assuming that high abundance OTUs that have low prevalence among samples from the same group are not relevant to characterise a group of samples. It uses random forests to classify groups of samples using high prevalence OTUs. Differences among groups were tested by the Kruskal–Wallis test or PERMANOVA based on the Bray–Curtis dissimilarity matrix. Differences were considered to be statistically significant at the level of *p* < 0.05.

## 3. Results

A total of 87 samples were obtained. After the exclusion of samples with an insufficient number of DNA sequences and samples with Good’s coverage value <0.85, fifty-eight samples were included in the analysis. The mean age of subjects was 65.2 (±13.8) years. 

Men aged 75 years and older had less diverse UM as measured by iChao2 (*p* = 0.035). Diabetic patients’ UM was less diverse than that of non-diabetics (*p* = 0.021, *p* = 0.028, *p* = 0.029, *p* = 0.033 and *p* = 0.017 for OTUs, ACE, iChao2, Shannon and Simpson indices, respectively) and subjects with elevated cholesterol and/or hyperlipidemia had less diverse microbiota than those with normal plasmatic lipid levels (*p* = 0.012 and *p* = 0.013 for ACE and iChao2, respectively). Current smokers‘ UM was more diverse than that of non-smokers by the Simpson index (*p* = 0.038). Study participants who were administered single-dose preoperative antibiotics had less rich UM than those without antibiotic prophylaxis (*p* = 0.001 and *p* = 0.037 for OTUs and iChao2, respectively). Please refer to Table 1 and Appendix A for details.

No statistically significant differences in any of the alpha diversity measures were detected for hypertension, CKD, the presence of ureteric stents or post-void residual volume of urine, “no growth” vs. “insignificant growth” on preoperative urine culture and BMI categories, International Prostate Symptom Score (IPSS) and International Index of Erectile Function (IIEF-5) score (Table 1 and Table 2, Figure 1 and Appendix A).

In terms of microbial community composition (beta diversity), significant differences were detected in microbiome composition between patients with CKD versus no CKD (*p* = 0.007), mild versus severe LUTS assessed by IPSS (*p* = 0.041) and between those who were administered single-dose antibiotics versus those with no prophylaxis (*p* = 0.001) (see Figure 2 and Table 3 for details).

## 4. Discussion

While correlations between the gut microbiota and certain pathological conditions have been demonstrated [2], this is less the case in the field of urobiome research, which lags behind that of the gut microbiota. The exact composition and importance of human UM is still a matter of debate; its functions might include competition with pathogens for resources, production of antimicrobial compounds or degradation of harmful substances, maintenance of epithelial junctions and even production of neurotransmitters [36].

Human gut microbiota has been shown to change with aging [37] but no dedicated, adequately powered study focused on age-related changes of the UM has been published to date. Lewis et al. detected certain genera only in the UM of the elderly (75 years and older) but their study population included only 16 subjects (six of which were males) [18]. Kramer et al. failed to detect differences in alpha diversity across age groups in their cohort study (n = 77) [19]. Our data indicate that men aged 75 years and older have a lower UM richness than their younger counterparts (*p* = 0.034 for iChao2). Female UM richness was associated with hormonal status [38] but unlike during menopause no abrupt decrease in circulating levels of sexual hormones occurs in men [39]. Hormonal changes alone are therefore unlikely to explain the decreased UM diversity; structural and functional changes of the mucosal surfaces might be responsible for the different compositions of UM in the elderly [40]. Interestingly, the composition of the gut microbiota of elderly people differs from that of young and middle-aged adults and this shift was reported to occur after 70 years of age. It is characterised by a decrease in microbial diversity, the accumulation of proinflammatory species and a cascade of inflammatory events leading to the impairment of the intestinal barrier integrity and subsequent morbidities, a process known as inflammaging [41]. With regard to the similar cut-off age for detectable differences between young and old in the gut and urinary tract, respectively, it is plausible that inflammaging occurs in the bladder as well as in the bowel.

The interplay of gut microbiota with the endocrine, nervous and immune systems co-regulates host blood pressure and kidney function via the (brain-)gut-kidney axis [42]. The gut microbiota in hypertensive rats as well as in humans exhibits lower richness and diversity than in normotensive individuals [43,44]. Similarly, UM of hypertensive subjects in our study showed a trend towards a lower UM richness and diversity. Two smaller reports of UM that tested, among other things, the putative link between hypertension and UM did not report any significant differences [8,19]. 

Diabetes mellitus has been linked to changes in gut microbiota composition [45]. We have tested the hypothesis that increased urinary glucose levels may promote bacterial growth and influence UM diversity and/or composition [19]. A study of 140 Chinese women with and without DM reported a decreased UM diversity in diabetic subjects versus controls and identified certain OTUs with differential relative abundances in either group [46]. Our data extend this finding to the male sex as all five alpha diversity indices were statistically significantly lower in diabetics in contrast to non-diabetics (Table 1). Enrichment of specific taxa was not demonstrated in either group. 

UM diversity has been reported to drop with an increasing degree of CKD [19]. This may be due to decreased uromodulin excretion by the renal tubules as eGFR declines. Although we did not demonstrate any significant differences in alpha diversity measures in our study population, beta diversity analysis using PIME identified 22 OTUs that differentiated patients with CKD from those without CKD (PERMANOVA r^2^ = 0.054, *p* = 0.007); Table 3, Figure 2A.

Activation of the renin-angiotensin system with dyslipidemia synergistically accelerates tubulointerstitial injury in mice. In diabetic nephropathy, tubulointerstitial injury is associated with the disruption of cholesterol homeostasis in kidney cells which in turn exacerbates diabetic nephropathy [47]. This mechanism described in diabetic rats might explain the significant difference in alpha diversity measures between subjects with dyslipidemia versus those without dyslipidemia (Table 1). Taken together, diabetes [46], chronic kidney disease [19] and dyslipidemia all seem to be associated with decreased UM diversity. Whether they act in a synergistic manner remains to be determined.

Increased UM diversity was associated with higher BMI in women [38]. Hence, we explored if BMI influences male UM. Neither our study nor two other studies [8,19] investigating this putative association as their secondary endpoint detected any link between UM and BMI in men (Table 2).

We observed increased UM diversity in smokers versus non-smokers (Simpson index, *p* = 0.038). This is in line with a previous report of smoking and non-smoking bladder cancer patients [48]. Another study failed to detect any differences in UM of smoking and non-smoking individuals [49].

Within 24 h after ureteric stenting, biofilm is formed on the stent surface and 90% of stents harbour microorganisms detectable by standard culture techniques [50]. We hypothesized that microbial communities in patients with stents would differ from those without due to significant overgrowth of some bacterial taxa on the stent surfaces and/or encrustations. However, no differences in alpha or beta diversity were observed. (NB: stented subjects were excluded from all other analyses in this work). We further hypothesized that post-void residual urine might influence the UM by a mechanism similar to the one postulated for ureteric stents. Again, no significant differences were observed between patients with post-void residual urine and those without.

We wanted to test whether UM differs in subjects with strictly negative urine culture reported as “no growth” by the clinical laboratory versus those reported as “contamination” or “mixed growth” up to 10^3^ colony forming units (CFU) mL^−1^. Sequencing of the 16S rRNA gene is able to describe more complex bacterial communities than the currently available culture techniques; unsurprisingly, no differences were detected between the two categories. Urine samples with no predominant organisms at 10^3^ CFU mL^−1^ may provide an unbiased reflection of the UM when subjected to sequencing.

In a study of lower urinary tract symptoms (LUTS) in males undergoing surgery for LUTS/benign prostate hyperplasia, the odds of detecting bacteria in catheterised urine samples using sequencing and EQUC (in a combined fashion) were associated with the degree of bother evaluated by the IPSS (22%, 30% and 57% in mild, moderate and severe bother, respectively). No specific genera were associated with the degree of LUTS [8]. We investigated potential correlations between IPSS bother groups and alpha and beta diversity. None were detected for the former (Figure 1A) but in the latter case, PIME detected certain OTUs to be enriched in mild versus severe bother and vice versa (PERMANOVA r^2^ = 0.083, *p* = 0.041; Figure 2B). A significant difference was detected between IPSS moderate bother versus mild and severe bother; because this occurrence is biologically implausible, we consider this a type I error (Table 2 and Figure 1).

Based on the same reasoning, we investigated potential differences in microbial community composition in men with different degrees of erectile dysfunction, as measured by the internationally validated IIEF-5 questionnaire. Measures of alpha diversity did not show any discernible trend, as one would expect for an ordinary variable (Table 2 and Figure 1B) and beta diversity analyses also ended with null results. A larger patient sample might provide better insight into the potentially important relationship between UM and erectile dysfunction. Importantly, confounding variables with known influence on the erectile function (age, hypertension, diabetes, cholesterol level, etc.) would have to be accounted for, allowing for unbiased conclusions.

Twenty-seven subjects enrolled in the study received a single pre-operative dose of antibiotics based on surgeon preference. Surprisingly, their UM richness was significantly lower than in those with no antibiotic prophylaxis (iChao2 and ACE *p* < 0.05). NMDS detected significant differences in the microbial communities between the two groups with *Pseudomonas*, *Brevundimonas* and *Burkholderia-Caballeronia-Paraburkholderia* enriched in individuals without prophylaxis and *Negativicoccus*, *Psychromonas* and *Propionimicrobium* enriched in the antibiotic group (see Table 1 and Figure 2C). Considering only a single dose of gentamicin was administered 30 min before the intervention, it is unlikely that the entire genera of microbes would be eradicated in such a short time. Even if so, their 16s rDNA should remain in urine and it should be detected with NGS. We offer two explanations for this unexpected finding: some, yet undefined, effect of antibiotics excreted in urine on the efficiency of DNA extraction; or the fact that patients who received antibiotic prophylaxis were more often diabetics (*p* = 0.003) and older (on average 68.9 and 61.9 years, respectively, *p* = 0.027) than those without antibiotic coverage. These factors, each associated with decreased microbiota diversity in our dataset, might have added up to create a false association between the administration of preoperative antibiotics and the composition of the UM. There were no significant differences between log_10_ total DNA concentration or log_10_ bacterial 16S rDNA concentration after DNA extraction between patients who received antibiotic prophylaxis versus those who did not. Due to the effect single-dose, antibiotic administration had on the UM and despite the fact that its underlying mechanism remains unclear, we suggest that the use of antibiotics prior to urine sampling be avoided in future UM studies.

Limitations of the present study include patient history as a source of information on diabetic status, hypertension, etc., instead of defined numerical parameters; smoking status was binary and did not reflect the extent of tobacco exposure. This study is one of the few that relies solely on catheterised urine samples. This is at the cost of recruiting control subjects among urological patients rather than healthy individuals. 

From the present study and other published evidence, it appears that in the majority of cases individuals with pathological conditions have a less diverse microbiota than individuals in whom these conditions are absent. Our results could serve as a starting point for future UM research which may re-test the described patterns in the microbial composition of urine, the direction of associations, and test cause-effect relationships between microorganisms and patients’ health status. Should the urinary microbiota composition represent a consequence of a given clinical situation, it might serve as a biomarker. Should the microbiota emerge as a cause of a pathological state then its therapeutic modification may also modify the resulting clinical condition.

Further research on UM would also benefit from larger patient populations, parallel testing of identical samples using different methods and from standardised reporting of results.

## 5. Conclusions

In our study, men aged 75 years and older had less rich UM than men younger than 75. Diabetes and dyslipidemia were associated with decreased UM richness and diversity. Preoperative administration of single-dose antibiotics was associated with decreased UM richness. Microbial community composition was significantly different in subjects with CKD versus those with normal kidney function; in men with IPSS mild versus severe bother; and in subjects who received antibiotic prophylaxis versus those who did not. Urine samples reported as “mixed growth” at 10^3^ CFU mL^−1^ seem to be comparable to those reported as “no growth” for the purposes of urinary microbiome research. Antibiotic administration prior to urine sampling for NGS should be avoided.

## Figures and Tables

**Figure 1 microorganisms-10-00874-f001:**
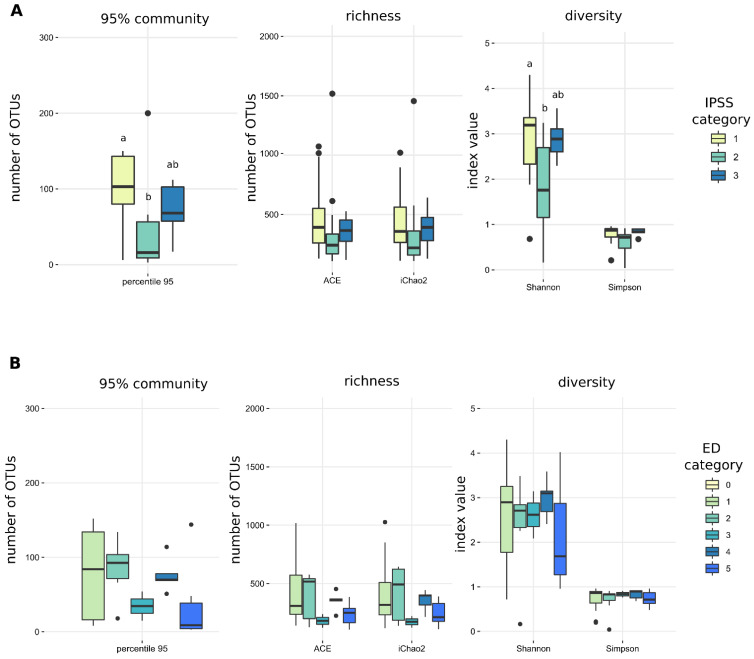
There were no discernible trends in alpha diversity measures for (**A**) International prostate symptom score (IPSS category) and (**B**) International Index of Erectile Function (ED category) despite a statistically significant difference in the number of OTUs and Shannon index between IPSS category 1 and 2 (see text for further details).

**Figure 2 microorganisms-10-00874-f002:**
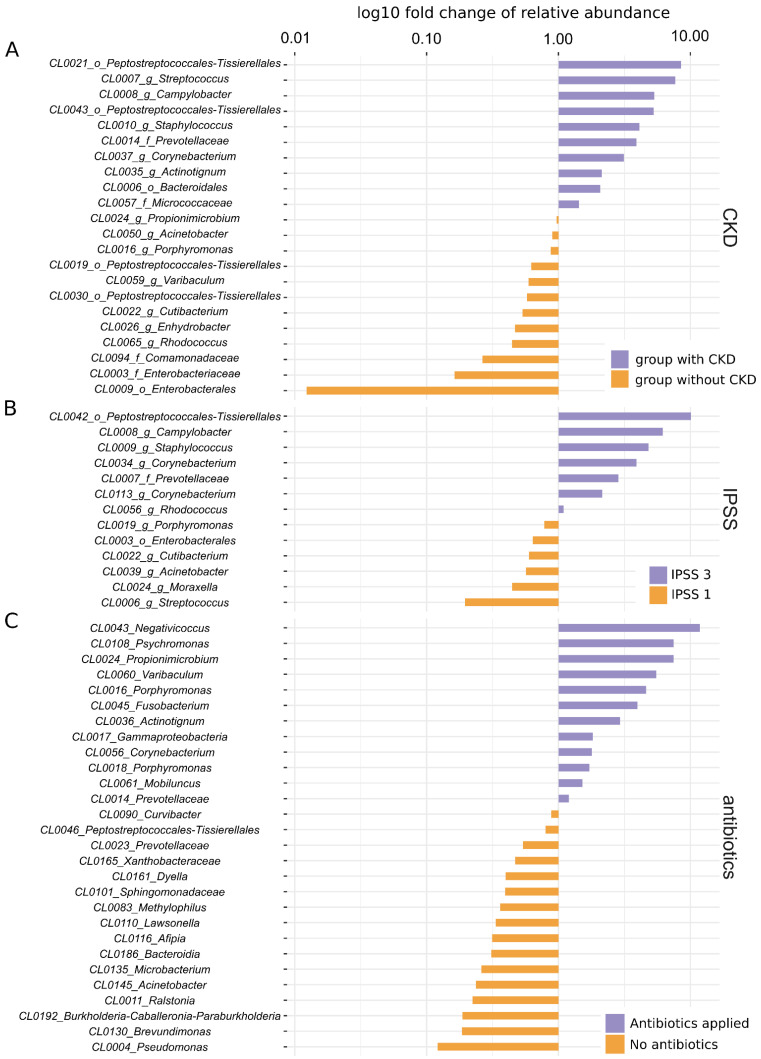
Key OTUs and their distribution in individual groups of patients. (**A**) Key OTUs for patients with CKD and without CKD. (**B**) Key OTUs for patients with IPSS 3 and IPSS 1. (**C**) Key OTUs for patients with and without antibiotic prophylaxis. Names of OTUs denote ID of the OTU together with the nearest assigned taxon, *X*-axis shows fold change of relative abundance of the taxa, enrichment in an individual group is colour-coded; values were log10 transformed. Note that the PIME algorithm puts emphasis on the prevalence of individual OTUs rather than their mean relative abundance; even the OTUs with the largest differential abundance between two groups may in fact have very low relative abundances.

**Table 1 microorganisms-10-00874-t001:** Median alpha diversity indices for the clinical conditions reported in this study. Columns denote the patient’s affiliation to individual clinical conditions (rows) and *p*-values. Values in boldface are statistically significant. OTUs—number of OTUs representing 95% of the community. Note the consistent differences across all alpha diversity measures for diabetes. See the Discussion section for more detail.

	n		OTUs			ACE			iChao2			Shannon			Simpson		
	No	Yes	No	Yes	*p*	No	Yes	*p*	No	Yes	*p*	No	Yes	*p*	No	Yes	*p*
Age (≥75 years)	34	16	89	34	0.057	316.5	221.9	0.052	**344.4**	**191.5**	**0.035**	2.59	2.52	0.507	0.79	0.80	0.602
Hypertension	19	31	92	49	0.100	347.1	244.0	0.132	348.6	250.0	0.090	3.00	2.39	0.054	0.86	0.78	0.105
Diabetes mellitus	36	14	**87**	**19**	**0.021**	**307.0**	**173.6**	**0.028**	**317.2**	**178.0**	**0.029**	**2.84**	**2.15**	**0.033**	**0.84**	**0.72**	**0.017**
Chronic kidney disease ^1^	36	10	64	35	0.390	306.2	302.4	0.485	315.7	301.6	0.466	2.53	2.39	0.976	0.79	0.81	0.585
Dyslipidemia	35	15	80	48	0.132	**307.8**	**193.0**	**0.012**	**344.4**	**185.9**	**0.013**	2.59	2.51	0.426	0.79	0.81	0.939
Current smoker	29	11	62	95	0.557	370.4	299.4	0.970	369.7	312.1	0.910	2.40	3.02	0.104	**0.77**	**0.85**	**0.038**
Ureteric stent	50	8	62	63	0.761	294.6	170.2	0.324	296.5	168.3	0.333	2.58	2.39	0.609	0.79	0.75	0.511
Post-void residual urine ^2^	30	4	81	80	0.953	356.8	392.4	0.814	348.4	382.4	0.953	2.67	2.40	0.289	0.84	0.78	0.239
Insignificant growth ^3^	38	12	80	19	0.101	312.1	218.7	0.090	327.2	209.5	0.196	2.67	2.31	0.544	0.83	0.78	1.000
Preoperative antibiotics	28	22	**97**	**19**	**0.001**	388.0	238.7	0.057	**388.1**	**252.2**	**0.037**	2.80	2.40	0.144	0.79	0.81	0.550

^1^ estimated glomerular filtration rate (CKD-EPI eGFR) <60 mL min^−1^. ^2^ 40 mL or more left in the bladder after spontaneous voiding as measured by ultrasound. ^3^ “Insignificant growth” at ≤10^3^ CFU mL^−1^ versus “no growth”.

**Table 2 microorganisms-10-00874-t002:** Median alpha diversity indices ± standard deviation for body mass index (BMI) categories: normal, overweight, obese; International Prostate Symptom Score (IPSS): mild, moderate and severe bother; and International Index of Erectile Function (IIEF-5): 1 = no or mild erectile dysfunction, 5 = significant erectile dysfunction. OTUs—number of OTUs representing 95% of the community. Note the U-shaped curve in the OTUs column for BMI and IPSS. Despite the statistical significance (*p* < 0.05), there is no discernible trend one would expect for an ordinary variable. We believe this to be due to sampling error.

	n	OTUs	*p*	ACE	*p*	iChao2	*p*	Shannon	*p*	Simpson	*p*
BMI											
<25.0	12	92.0 ± 54.4	0.402	397.5 ± 282.0	0.550	399.8 ± 230.2	0.586	2.5 ± 0.9	0.795	0.8 ± 0.2	0.753
25.0–29.9	23	39.5 ± 51.7		252.6 ± 342.3		266.5 ± 325.7		2.5 ± 0.6		0.8 ± 0.1	
≥30.0	13	113.0 ± 92.8		273.4 ± 283.7		284.4 ± 277.9		3.1 ± 1.5		0.9 ± 0.3	
IPSS											
1	17	103.0 ± 51.1	0.009	392.3 ± 311.0	0.205	357.1 ± 278.5	0.246	3.2 ± 0.9	0.034	0.9 ± 0.2	0.066
2	14	16.0 ± 55.5		241.8 ± 378.5		218.6 ± 361.7		1.8 ± 1.0		0.7 ± 0.3	
3	8	68.0 ± 34.6		365.0 ± 149.3		392.3 ± 176.2		2.9 ± 0.4		0.8 ± 0.1	
IIEF-5											
1	12	84.0 ± 59.7	0.190	308.4 ± 295.4	0.298	318.3 ± 279.3	0.294	2.9 ± 1.1	0.778	0.9 ± 0.3	0.857
2	8	92.5 ± 39.6		517.3 ± 200.4		493.3 ± 233.9		2.7 ± 1.1		0.8 ± 0.3	
3	3	34.5 ± 27.6		180.3 ± 83.2		172.1 ± 70.6		2.6 ± 0.7		0.8 ± 0.1	
4	5	70.0 ± 23.2		364.5 ± 82.3		398.0 ± 91.4		3.1 ± 0.5		0.9 ± 0.1	
5	7	9.0 ± 55.6		250.3 ± 102.3		213.0 ± 110.5		1.7 ± 1.2		0.7 ± 0.2	

**Table 3 microorganisms-10-00874-t003:** Summary of beta diversity analyses. Notably, there was a significant difference in microbiome composition between patients with CKD versus no CKD (*p* = 0.007), mild versus severe LUTS assessed by IPSS (*p* = 0.041) and between those who were administered single-dose antibiotics versus those with no prophylaxis (*p* = 0.001) (Figure 2).

	n	PIME OTUs	R^2^	PERMANOVA *p*
Age (≥75 years)	50	29	0.015	0.727
Hypertension	50	29	0.017	0.574
Diabetes mellitus	50	9	0.027	0.203
Chronic kidney disease	46	22	0.054	0.007
Dyslipidemia	50	29	0.021	0.364
Obesity	48	155	0.016	0.724
Current smoker	40	21	0.021	0.630
Ureteric stent	58	14	0.017	0.448
Post-void residual urine	34	29	0.018	0.806
Insignificant growth	50	29	0.019	0.547
IPSS (mild vs. severe)	39	13	0.083	0.041
IIEF-5	35	14	0.033	0.331
Preoperative antibiotics	50	29	0.062	0.001

n: number of samples with clinical information available entering the analysis; IPSS: International Prostate Symptom Score; IIEF-5: International Index of Erectile Function; PIME OTUs: number of OTUs classified by the Prevalence in Microbiome Analysis; algorithm as key for dissimilarities among groups based on their prevalence; R^2^: the proportion of total variability of the samples that is explained by the PIME OTUs.

## Data Availability

The DNA sequences have been deposited at the NCBI SRA (PRJNA744742).

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
