# Peer review of "Bladder Microbiota Are Associated with Clinical Conditions That Extend beyond the Urinary Tract"

_microorganisms, 2022, doi:10.3390/microorganisms10050874_

Round 1
Reviewer 1 Report
The authors seek to understand whether there are differences in the urinary microbiota of men with a variety of comorbidities or experiencing various clinical conditions. The authors methods are strong and they use the appropriate controls needed for low microbiome samples like the urinary tract. This study validates other publications and extends it to a new patient population, such as the association of increased microbial diversity in smokers vs nonsmokers. This study also provides insights into the urinary microbiome of men, which has received less attention than women. While this work adds important insights into the urinary microbiome of men, enthusiasm for the manuscript was diminished by the abbreviated intro and results sections. Additionally, several key issues should be address to strengthen the conclusions of the paper including the following:
Major:
Intro:
While the authors’ intro discusses what the urinary microbiome is and points out several clinical conditions that are known to be associated with changes in the diversity of the microbiome, this discussion is brief. Additionally, the discussion of alpha and beta diversity seems better suited to the materials and methods section. Instead, it would help orient the reader if the authors could expand their discussion of the urinary microbiome, specifically introducing the clinical conditions they assess in their work and what is known, if anything, about those conditions in relation to microbial diversity or in the male urinary tract compared to the female urinary tract, which has received far more attention.
Materials and Methods:
While the authors reference a previous publication regarding study population, it would provide clarity to the reader and greatly strengthen the paper if the authors could include minimum patient info such as: reason for endoscopy, particularly in reference to those undergoing surgery due to bladder cancer.
Results:
Line 121: Can the authors provide details as to why age 75 was chosen as the cutoff, as opposed to 65 or 85?
The authors note that almost half (46%) of the study population has bladder cancer, but no comparisons are made between those with and without cancer. Furthermore, there is almost no mention of cancer throughout the entire manuscript, with the exception of a brief mention of a previous report that indicated cancer was associated with “changes in the urinary microbiota”. However, what those changes are, are never mentioned. It would greatly strengthen the paper if the authors would include these same analyses for participants with and without cancer and compare their results to the previous report.
Figure 1 is not referenced in the results section at all. Please include a discussion of the figure, or indicated where in the results section the results from that figure are being discussed.
While the legends for Table 3 and Figure 2 are well written and describe the main points of the data, Tables 1 &2 and Figure 1 only describe the analyses that were performed. Can the authors include a description of the main take-a-ways for each panel/figure to help orient the reader? Additionally, can the authors give an explanation for the U shaped curve observed in the OTUs column for BMI in Table 2? It is unclear how both normal and obese BMI would have an observed diversity of 92 and 113 while the normal BMI range has an observed diversity of 39.5. Ranges of observations would help understand and interpret this entire table. These are present in Figure 2
Can the authors indicate if they included age or cancer status as potential confounders in their PERMANOVA models to see if some of the observations made are due to correlations with other risk factors? In particular, for Table 3, the authors note that no study to date has reported specifically on age-related changes to the urinary microbiome. This potential for confounding as an age effect is important to assess.
Discussion:
Line 173: “leels” should be “levels”
Lines 174-176: The authors hypothesize that changes in the male urinary microbiome may differ from females, since they don’t have large hormonal changes associated with menopause, and that changes they observe may be due to functional or structural changes within the bladder of the elderly. However, it would strengthen the authors conclusions if they could also briefly discuss what role inflammaging might have on the urinary microbiome of the elderly?
Lines 260-270: the authors show that subjects who received abx prophylaxis had different microbial communities than those who didn’t. They propose that abx may inhibit DNA extraction or there may be confounding comorbidities. Were the subjects who were excluded due to insufficient number of DNA sequences given antibiotics? Or can the authors comment on whether there might be another inhibitor in present in urine? Furthermore, could the authors also comment on the type of antibiotic administered? Was it broad-spectrum? Much of the communities outline in Figure 1, eg strep, staph, enterobacterales, would be susceptible to many of the broad-spectrum antibiotics typically used in urinary prophylaxis. Could the authors briefly discuss what affects that antibiotic may have in shifting the microbiota from potential susceptible microbes to ones that may be more resistant?
Lines 210-212: the authors mention that urinary microbial diversity increases in smokers vs nonsmokers and that a previous report in bladder cancer patients also found this result. Considering almost half of their subjects are cancer patients, have the authors included cancer status as potential confounders in their PERMANOVA models? Also a discussion of any comparisons performed among those with and without cancer would be helpful to the reader.
Lines 276-277: The authors state that “…most pathological conditions are associated with a less diverse microbiota than health.” But the authors study population does not include any “healthy” individuals. All participants are undergoing some sort of urological intervention. The authors should therefore include in the materials and methods or the results section which group they are considering “healthy/normal” among each of the different comparisons. The authors should also consider tempering their language in these lines to not mislead the reader in to thinking they include a “healthy” control group.
Minor:
Abstract:
Line 18: “The composition of [the] bacterial community in urine” the word “the” is missing
Materials and Methods:
Line 75: Can the authors explain why the V4 region was used for sequencing instead of V3 or even the V3-V4 Illumina Microbiome kit, which gives better resolution at the species level?
Line 94: what does EPI in CKD-EPI stand for?
Results:
Line 118: supplemental table 1 should be referenced when discussing the patient demographics
Reviewer 2 Report
The authors present an observational study on relationships between the urinary microbiome in men and various clinical parameters. They find numerous correlations, some of which are novel, and some of which support prior research. The study is well designed, and the manuscript is well written. The discussion section might be improved by considering the following comments.
I understand that this is a hypothesis generating study, but the authors should elaborate on the potential clinical implications of this line of research. Suppose that we conducted larger more extensive studies on this subject and obtained a near complete understanding of the correlation between the urinary microbiome and specific health conditions. How could we use that information to improve patient care?
In clinical practice, we sometimes recommend oral probiotics as a prophylactic measure for patients with recurrent UTIs. As far as I am aware, there are no clinical data correlating oral probiotic use to the urinary microbiome. This could be a future area of study with a clear potential clinical intervention.
Reviewer 3 Report
In this manuscript, the authors analyze the diversity of bladder microbiome and clinical indicators in patients. Although a multi-group analysis was carried out, no meaningful results were obtained.
(1)The urine of 87 male patients was collected by catheterization, but only 58 cases of 16s rRNA sequencing data were produced, possibly because urine storage conditions should be -80℃ rather than -20℃.
(2)According to the contents of the manuscript, 87 patients were assumed to be inpatients in urology department, and there were no other healthy controls.
(3)The results showed that diabetes, dyslipidemia and smoking were related to the change of urinary microbiota, but such conclusions have been drawn before (PMID: 28316574, 34168995).
(4)Studies of chronic kidney disease, lower urinary tract symptoms and urine microbiota have also been described (PMID: 34313465、29230385、30270128).
Round 2
Reviewer 3 Report
In general, this version of the manuscript corrects the errors of the previous , but it still falls short. Several comments are below:
The statistical tables in the manuscript should be standardized as in Table 3.
The authors should report the actual p-values, not >0.05 or <0.05 throughout the Results section.
The authors should read the Urobiome Consensus paper (Brubaker et al., 2021 PMID: 34282932) and added description of sample preservation such as whether DNA/RNA shield was used.
